# Associations between artificial sweetener intake from cereals, coffee, and tea and the risk of type 2 diabetes mellitus: A genetic correlation, mediation, and mendelian randomization analysis

**Youqian Zhang[1], Zitian Tang[2], Yong Shi[3], Lin Li[1]***

**1** Department of Endocrinology, The First Affiliated Hospital of Yangtze University, Jingzhou, Hubei, China,
**2** Department of Law, Yangtze University, Jingzhou, Hubei, China, **3** Department of Medicine, Yangtze University, Jingzhou, Hubei, China

\* 2022721046@yangtzeu.edu.cn

## Abstract

### Background

Previous studies have emphasized the association between the intake of artificial sweeteners (AS) and type 2 diabetes mellitus (T2DM), but the causative relationship remains ambiguous.

### Methods

This study employed univariate Mendelian randomization (MR) analysis to assess the causal link between AS intake from various sources and T2DM. Linkage disequilibrium score (LDSC) regression was used to evaluate the correlation between phenotypes. Multivariate and mediation MR were applied to investigate confounding factors and mediating effects. Data on AS intake from different sources (N = 64,949) were sourced from the UK Biobank, while T2DM data were derived from the DIAbetes Genetics Replication And Meta-analysis.The primary method adopted was inverse variance weighted (IVW), complemented by three validation techniques. Additionally, a series of sensitivity analyses were performed to evaluate pleiotropy and heterogeneity.

### Results

LDSC analysis unveiled a significant genetic correlation between AS intake from different sources and T2DM ($r_g$ range: -0.006 to 0.15, all $P$ < 0.05). After correction by the false discovery rate (FDR), the primary IVW method indicated that AS intake in coffee was a risk factor for T2DM (OR = 1.265, 95% CI: 1.035–1.545, $P$ = 0.021, $P_{FDR}$ = 0.042). Further multivariable and mediation MR analyses pinpointed high density lipoprotein-cholesterol (HDL-C) as mediating a portion of this causal relationship. In reverse MR analysis, significant evidence suggested a positive correlation between T2DM and AS intake in coffee (β =

**Data Availability Statement:** All relevant data are within the manuscript and its Supporting Information files.

**Funding:** The author(s) received no specific funding for this work.

**Competing interests:** The authors have declared that no competing interests exist.

0.013, 95% CI: 0.004–0.022, $P = 0.004$, $P_{FDR} = 0.012$), cereal (β = 0.007, 95% CI: 0.002–0.012, $P = 0.004$, $P_{FDR} = 0.012$), and tea (β = 0.009, 95% CI: 0.001–0.017, $P = 0.036$, $P_{FDR} = 0.049$). No other causal associations were identified ($P > 0.05$, $P_{FDR} > 0.05$).

## Conclusion

The MR analysis has established a causal relationship between AS intake in coffee and T2DM. The mediation by HDL-C emphasizes potential metabolic pathways underpinning these relationships

## Introduction

Diabetes mellitus (DM) represents a significant and pressing global health concern [1], with type 2 diabetes mellitus (T2DM) constituting approximately 90% of all diabetes cases worldwide [2]. The World Health Organization (WHO) estimates that there are currently over 422 million diabetics globally and that there will be 629 million by the year 2045 [3, 4]. Notably, the prevalence of diabetes has experienced an upward trajectory in developing nations, including China and Pakistan, leading to considerable direct and indirect financial strain on society [5]. Consequently, the identification of novel modifiable risk factors for T2DM is imperative for informing clinical management strategies and mitigating the onset and progression of the disease.

As lifestyles change, the demand for sweet treats is gradually increasing. Artificial sweeteners (AS), as low-calorie and sugar-free alternatives, have gained popularity as sugar substitutes to decrease caloric intake [6]. The most popular AS include aspartame, saccharin, acesulfame potassium, and sucralose [7], commonly used in foods and beverages such as cereals [8], coffee [9], and tea [10] to satisfy the demand for sweetness. Current research has identified associations between AS and T2DM; however, findings from observational studies in this domain often exhibit inconsistencies. Certain investigations have reported a 3% elevated relative risk of T2DM per additional daily serving of AS [11–14], while others have demonstrated that the intake of artificially sweetened beverages, when compared to water, is associated with a 21% rise in T2DM incidence [15]. Moreover, no correlation between AS and T2DM has been shown in other studies [16, 17]. Despite the widespread use of AS in the daily diet and their popularity among people with T2DM, there is no consensus on a causal relationship between them and diabetes due to inconsistent research findings.

Previous research encountered challenges in establishing a definitive causal relationship between exposure factors and outcome variables, largely attributable to complexities stemming from confounding variables and reverse causation. Given the constraints of observational studies in ascertaining causality with certainty, alternative approaches such as Mendelian randomization (MR) in the realm of genetic research can prove to be invaluable. Experiments that employ MR utilize genetic variations, ascertained through genome-wide association analyses, as instrumental variables (IVs). These IVs help in gauging the causal relationship between environmental exposure and the desired outcome. Under certain conditions, this technique allows for drawing causal inferences by using genetic variants as surrogates for environmental exposure [18]. Conceived as a natural randomized controlled trial, MR is based on Mendelian inheritance laws that allocate parental alleles to their offspring. This approach offers a more robust degree of evidence and a diminished vulnerability to confounding factors. In contrast to observational epidemiological research, MR presents a higher caliber of evidence. This

study aims to employ univariate MR (UVMR), multivariate MR (MVMR), mediation MR, and linkage disequilibrium score (LDSC) regression to investigate the relationship between intake of AS from various sources and T2DM, further delving into the mediating roles of five confounding factors.

## Materials and methods

### Study design

The foundational datasets for this study were procured from genome-wide association studies (GWAS). Each GWAS study included obtained the necessary approvals from their respective institutional review boards. As this study involves a secondary analysis of publicly available data, no additional ethical permissions were required. IVs for the exposure were identified based on three critical criteria: (i) the selected genetic variant, serving as the IV, should have a strong association with the exposure; (ii) this variant should not be associated with any known confounders; and (iii) the effect of the variant on the outcome should be solely through the exposure, negating any alternative pathways [19]. The MR approach is detailed in **Fig 1**, while the summary statistics from the data sources are presented in **Table 1**.

### Selection of genetic instrumental variables

The summary-level GWAS data for AS intake in coffee/tea/cereal were all sourced from the UK Biobank (UKB) [20], encompassing 64,949 European individuals. This information was collected using questionnaires where participants detailed the amount of AS (for example, Canderel) they added to their daily coffee or tea/infusion on a per-drink basis. Additionally, those who reported consuming cereal or porridge the previous day specified the quantity of sweetener added per bowl. To ensure the accuracy of MR analyses, we adhered to stringent criteria for single nucleotide polymorphism (SNPs) selection: (i) SNPs selected as IVs must show an association with the defined exposure at a genome-wide significance level ($P < 5 \times 10^{-8}$). Given the absence of genome-wide significant SNPs for exposure, we applied a relaxed threshold of $5 \times 10^{-6}$ to capture more SNPs for these phenotypes [21]. (ii) Chosen SNPs were further filtered to ensure no associations with potential confounders and to preserve independence among them, thereby mitigating potential biases from linkage disequilibrium ($r^2 < 0.001$, clumping distance = 10,000 kb). (iii) The efficacy of the selected SNPs as IVs was validated using F-statistics ($F = beta^2/se^2$; beta for SNP-exposure association; variance (se)), assessing the possibility of weak instrumental variables [22]. A high F-statistic indicates robust instrumental strength, and our criteria required all SNPs to have an F-statistic above 10. (iv) To enhance the reliability of our results, we applied MR-Steiger filtering, which systematically excludes variants more correlated with outcomes than exposures [23]. (v) In cases where an SNP is absent from the outcome dataset, we utilized the SNiPa online platform (http://snipa.helmholtz-muenchen.de/snipa3/), based on European population genotype data from the 1000 Genomes Project's Phase 3, to locate the missing SNP and identify a proxy SNP with strong linkage disequilibrium (criteria set at $r^2 > 0.8$) to the original SNP. (vi) The effect of the SNP on the exposure and its effect on the outcome should align with the same allele.

### Source of outcome phenotypes

The summary-level GWAS meta-analysis for T2DM integrated 22 cohorts, sourced from the AMP-T2D Knowledge Portal and the DIAbetes Genetics Replication And Meta-analysis (DIA-GRAM) consortium [24]. T2DM is defined by ICD-10 codes and includes 180,834 cases and 492,191 controls from European populations.

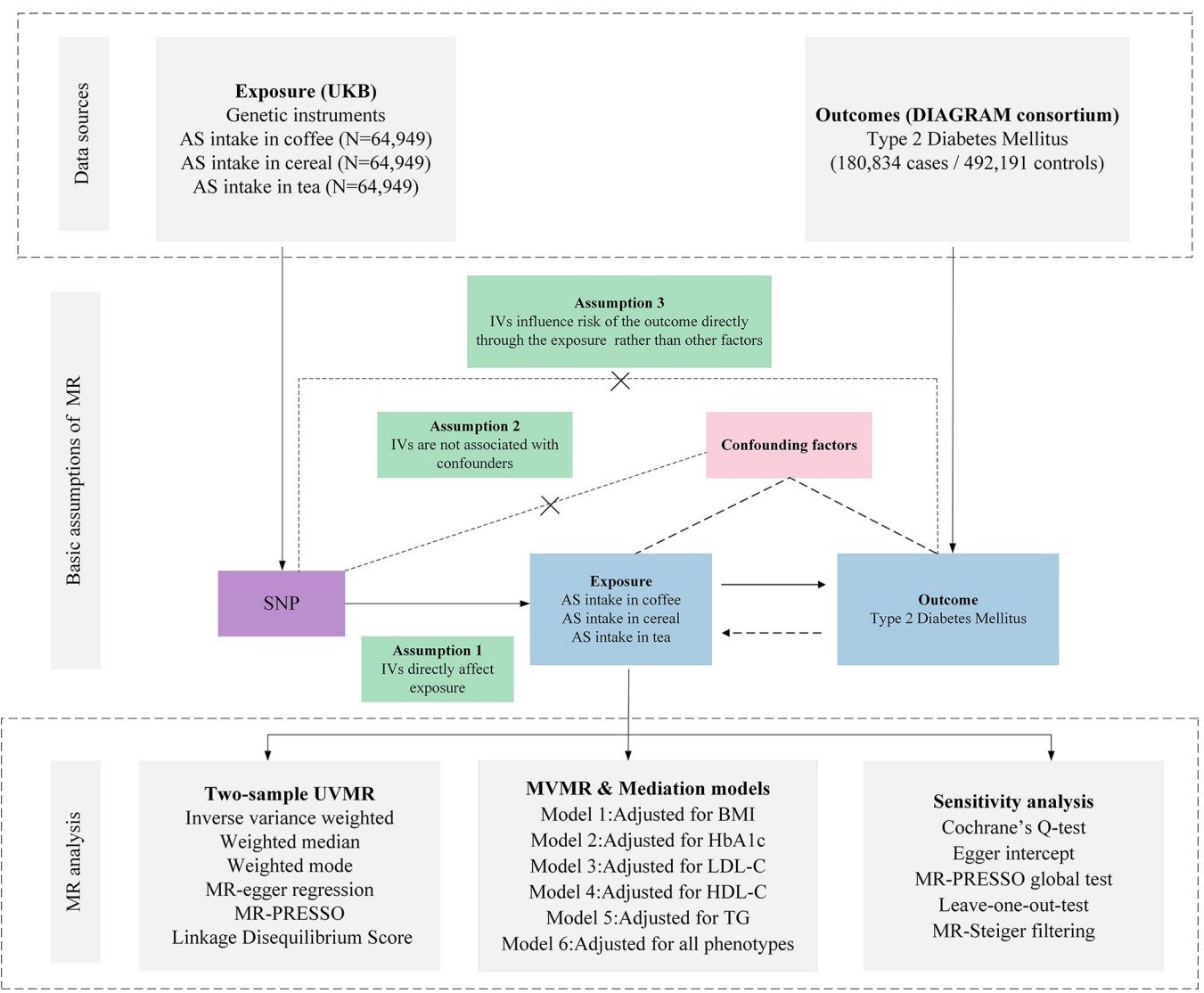

**Fig 1. Overview of research design and analysis strategy.** Overview of the research design. Exposures come from UKB, with outcomes including Type 2 diabetes mellitus. The MR framework is based on three fundamental MR assumptions, with MVMR analyses adjusting for five mediating factors for positive results. MVMR, Multivariate MR; UVMR, Univariate MR; BMI: Body Mass Index; SNP, Single Nucleotide Polymorphism; MR-PRESSO, MR Pleiotropy Residual Sum and Outlier; HbA1c, Glycated Hemoglobin A1c; LDL-C, Low Density Lipoprotein Cholesterol; HDL-C, High Density Lipoprotein Cholesterol; TG, Triglyceride; DIAGRAM, DIAbetes Genetics Replication And Meta-analysis; UKB, UK Biobank; AS, Artificial sweetener.

## Data sources for possible mediators

We further obtained genetic associations for Body Mass Index (BMI) from the Genetic Investigation of Anthropometric Traits (GIANT) consortium [25], Glycated Hemoglobin A1c (HbA1c) from Meta-Analyses of Glucose and Insulin-related traits Consortium (MAGIC) [26], triglycerides (TG), Low-Density Lipoprotein Cholesterol (LDL-C) and High-Density Lipoprotein Cholesterol (HDL-C) from Global Lipids Genetics Consortium (GLGC) [27].

## Statistical analyses

### Primary MR analysis

Within the UVMR framework, individual IVs were evaluated using the Wald ratio test. For scenarios involving multiple IVs, the multiplicative random-effects inverse-variance-weighted

**Table 1. Detailed information of data sources.**

| Explore or Outcome | Ref | Ieu id | Consortium | Ancestry | Participants |
|---|---|---|---|---|---|
| **Phenotypes** | | | | | |
| AS intake in coffee | 36402876 | ukb-b-1338 | UKB | European | 64,949 individuals |
| AS intake in cereal | 36402876 | ukb-b-3143 | UKB | European | 64,949 individuals |
| AS intake in tea | 36402876 | ukb-b-5867 | UKB | European | 64,949 individuals |
| T2DM | 35551307 | NA | DIAGRAM | European | 180,834 cases / 492,191 controls |
| **Adjustment of the model** | | | | | |
| LDL-C | 24097068 | ieu-a-300 | GLGC | 96% European | 173,082 individuals |
| HDL-C | 24097068 | ieu-a-299 | GLGC | 96% European | 187,167 individuals |
| TG | 24097068 | ieu-a-302 | GLGC | 96% European | 177,861 individuals |
| HbA1c | 20858683 | ieu-b-104 | MAGIC | European | 46,368 individuals |
| BMI | 30124842 | ieu-b-40 | GIANT | European | 681,275 individuals |

DIAGRAM, DIAbetes Genetics Replication And Meta-analysis; T2DM, type 2 Diabetes Mellitus; UKB, UK Biobank; BMI, body mass index; GIANT, Genetic Investigation of Anthropometric Traits; GLGC, Global Lipids Genetics Consortium; MAGIC, Meta-Analyses of Glucose and Insulin-related traits Consortium; LDL-C, Low Density Lipoprotein Cholesterol; HDL-C, high density lipoprotein-cholesterol; TG, triglyceride; HbA1c, glycated hemoglobin A1c; Ref, reference(Pubmed id); AS, artificial sweetener.

(IVW) approach was utilized, with ancillary use of the MR-Egger and weighted median methodologies. In the IVW approach, the weight accorded aligns directly with the Wald ratio estimation and inversely with the variance of each respective SNP [28]. IVW offers dependable outcomes when all genetic variants are appropriate; however, the weighted median is optimal when a majority are deemed inappropriate, and MR-Egger is reserved for complete invalidity [29]. Moreover, we utilized the constrained maximum likelihood (CML) method for our analysis. This technique allows for the combined estimation across multiple genetic variants while considering possible confounders and genetic heterogeneity. Using CML ensures enhanced precision and reliability in our estimates, especially in scenarios involving an abundance of genetic variants and potential confounding variables [30]. Consideration for multiple comparisons was made through the false discovery rate (FDR), with a post-adjustment $P$-value $< 0.05$ indicating a discernible causal association. Situations where a raw $P$-value was $< 0.05$, yet exceeded 0.05 post-FDR adjustment, were considered indicative rather than conclusive.

Within the scope of the Mediation MR and MVMR analysis, and recognizing potential confounders such as BMI, HbA1c, TG, LDL-C, and HDL-C in the exposure-outcome trajectory, we employed MVMR to discern the inherent causal relationship. The initial MVMR postulation centers on the association of genetic variations with specific exposures, while subsequent postulates align with UVMR standards [31]. An assessment was conducted to quantify mediated effects. Commencing with MR, effect estimates correlated to exposures were ascertained via the IVW approach. Subsequently, MVMR was utilized to quantify the influence of the aforementioned mediating factors on outcomes. The product of the two estimates for each outcome yielded the exposure's indirect influence. The ratio of the mediated to the total effect facilitated an understanding of each mediator's contribution to the cumulative effect.

## Genetic correlation analysis

Linkage disequilibrium score (LDSC) regression, when tailored to summary-level GWAS datasets, emerges as a sophisticated technique to interrogate genetic correlations in intricate diseases or varied phenotypic manifestations. With precision, this methodology segregates authentic polygenic associations from potential confounders, such as cryptic relatedness or population

stratification [32]. A statistically and quantitatively significant genetic correlation implies that the aggregate phenotype correlation extends beyond mere environmental confounders [32]. For detailed investigations into genetic correlations between specific exposures and corresponding phenotypic outcomes, the LDSC platform is available at https://github.com/bulik/ldsc.

### Sensitivity analysis

Within the UVMR structure, we executed a series of tests to validate the analytical integrity. The Cochran's Q test was utilized to measure heterogeneity among the chosen genetic markers, recognizing a *P*-value under 0.05 as indicative of significant variances between the SNPs under examination [33]. The MR-Egger regression was employed to explore the potential of directional pleiotropy in the MR framework [34]. An intercept *P*-value below 0.05 in the MR-Egger regression indicates a considerable directional pleiotropy, albeit the method may pose limitations in accuracy [35]. The MR Pleiotropy Residual Sum and Outlier (MR-PRESSO) was invoked to pinpoint outliers and assess horizontal pleiotropy, deeming a global *P*-value under 0.05 as significant [36]. This technique's exclusion of outliers fine-tunes the correction process. Additionally, the leave-one-out method was incorporated to evaluate the influence of individual SNPs on the overall results [37].

The $R^2$ value was computed using the formula $2 \times MAF \times (1-MAF) \times beta^2$, with MAF representing the minor allele frequency of each instrumental SNP. Summation of these values yielded the coefficient pivotal for power computation [38]. We derived the statistical power utilizing tools available on the mRnd website [39] (https://shiny.cnsgenomics.com/mRnd/).

## Results

### Genetic instrument selection and genetic correlation between phenotypes

The study reports F statistics exceeding 20 for all IVs, signifying a robust reduction of bias from weak instruments. The SNPs selected as IVs ranged from 14 to 176, accounting for an explained variance of 0.09% to 28.47%, and the power statistics obtained ranged from 6% to 100% (S1 Table).

LDSC analysis revealed a significant genetic correlation between AS intake in coffee ($r_g$: 0.384, $P = 1.19 \times 10^{-8}$), AS intake in tea ($r_g$: 0.263, $P = 3.70 \times 10^{-3}$), and AS intake in cereal ($r_g$: 0.932, $P = 8.12 \times 10^{-7}$) with T2DM. The SNP-based heritability ($h^2$) was 1.42% (coffee), 0.24% (tea), 1.43% (cereal) and 5.71% (T2DM).

### Association of genetically predicted exposure with outcome

The scatter plot provides a clear visualization of all positive results (Fig 2). After applying the FDR correction for multiple comparisons, our primary IVW analysis demonstrated strong causal evidence (Fig 3). Specifically, for every standard deviation (SD) increase in genetically predicted AS intake in coffee, the risk of T2DM increased by 26% (OR = 1.265, 95% CI 1.035~1.545, $P = 0.021$, $P_{FDR} = 0.042$). Complementary methods IVW-fixed effects (OR = 1.265, 95% CI 1.042~1.535, $P = 0.018$, $P_{FDR} = 0.036$)and CML (OR = 1.276, 95% CI 1.035~1.574, $P = 0.022$, $P_{FDR} = 0.044$) provide evidence of consistency. With an OR of 1.265, we possess 98% statistical power to detect an association between AS intake in coffee and T2DM.Furthermore, no causal association was identified between AS intake in cereal and T2DM (OR = 1.020, 95% CI 0.755~1.378, $P = 0.897$, $P_{FDR} = 0.897$), and similarly, for AS intake in tea (OR = 1.174, 95% CI 0.933~1.477, $P = 0.172$, $P_{FDR} = 0.206$) (S2 Table). With OR of 1.020 and 1.174, we have 6% and 77% statistical power, respectively, to detect associations between AS intake in cereal and tea and T2DM.

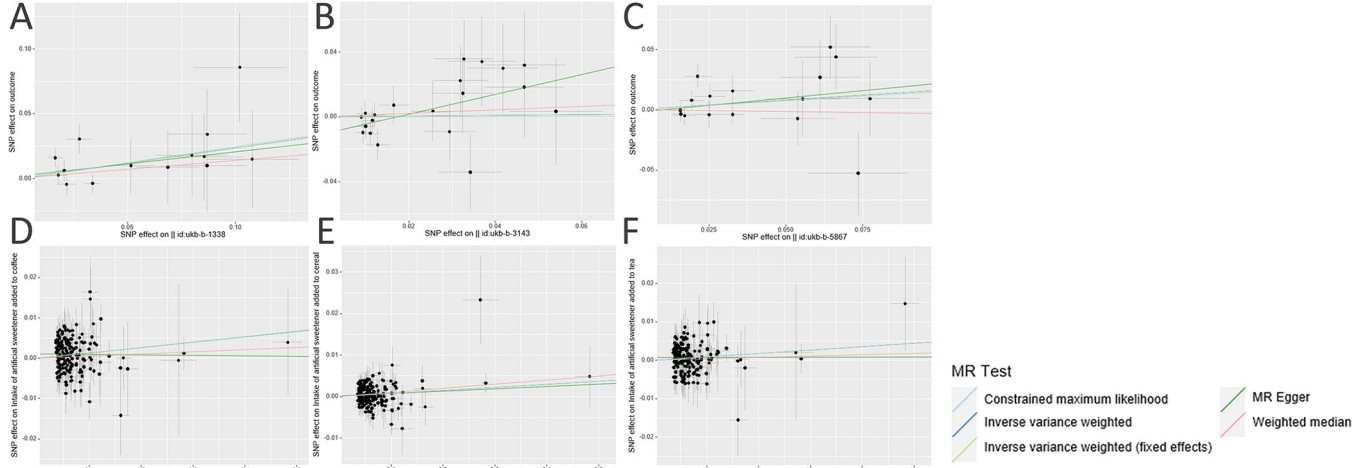

**Fig 2. Genetic associations with AS intake from different sources (horizontal axis, standard deviation units) and with T2DM (vertical axis, log odds ratios) at a genome- wide level of significance.** (A) AS intake in coffee on T2DM (B) AS intake in tea on T2DM (C) AS intake in cereal on T2DM (D) T2DM on AS intake in coffee (E) T2DM on AS intake in tea (F) T2DM on AS intake in cereal. Horizontal and vertical lines represent 95% confidence intervals for the genetic associations. Horizontal and vertical lines represent 95% confidence intervals for the genetic associations. AS, Artificial sweetener; T2DM, type 2 diabetes mellitus.

In the reverse MR analysis utilizing the IVW method (**Fig 4**), there is compelling evidence suggesting a positive causal relationship between T2DM and AS intake in coffee (β = 0.013, 95% CI 0.004~0.022, *P* = 0.004, $P_{FDR}$ = 0.012), cereal (β = 0.007, 95% CI 0.002~0.012, *P* = 0.004, $P_{FDR}$ = 0.012), and tea (β = 0.009, 95% CI 0.001~0.017, *P* = 0.036, $P_{FDR}$ = 0.049) (**S2 Table**). With β of 0.013, 0.007, and 0.009, we have 100% statistical power to detect associations between T2DM and AS intake in cereal, tea, and coffee, respectively.

In UVMR, the study identified AS intake in coffee with evidence for a causal relationship with T2DM that passed their statistical significance threshold ($P < 0.05$ & $P_{FDR} < 0.05$), and neither of these associations were significant in MVMR analyses that accounted for potential confounders (LDL-C, HDL-C, TG, HbA1c, BMI and all models) (**S3 Table**).Further mediation MR analysis revealed that HDL-C partially mediates the causal relationship with AS intake in

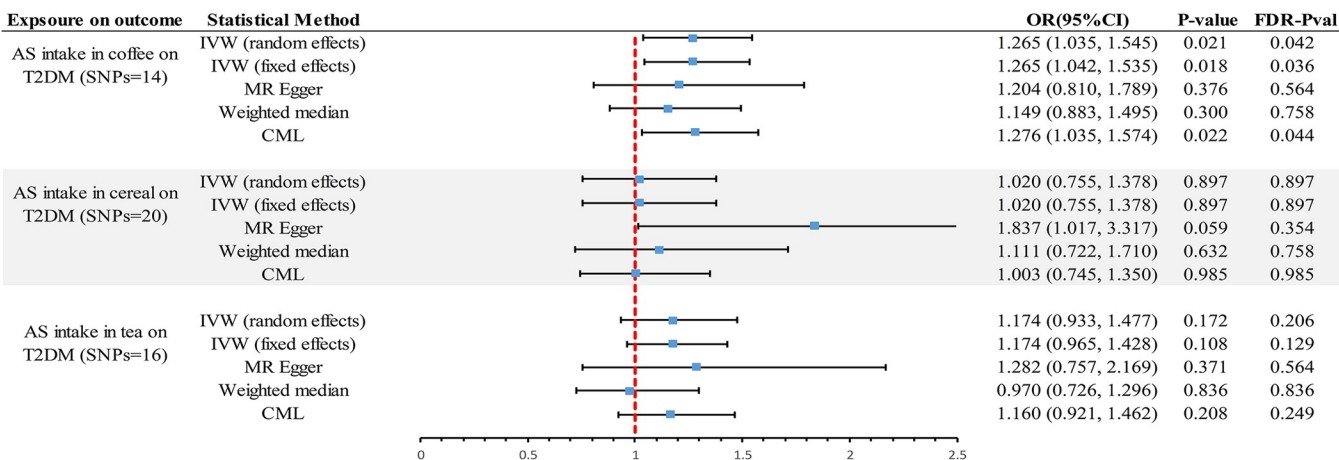

| Expsoure on outcome | Statistical Method | | OR(95%CI) | P-value | FDR-Pval |
|---|---|---|---|---|---|
| AS intake in coffee on T2DM (SNPs=14) | IVW (random effects) | | 1.265 (1.035, 1.545) | 0.021 | 0.042 |
| | IVW (fixed effects) | | 1.265 (1.042, 1.535) | 0.018 | 0.036 |
| | MR Egger | | 1.204 (0.810, 1.789) | 0.376 | 0.564 |
| | Weighted median | | 1.149 (0.883, 1.495) | 0.300 | 0.758 |
| | CML | | 1.276 (1.035, 1.574) | 0.022 | 0.044 |
| AS intake in cereal on T2DM (SNPs=20) | IVW (random effects) | | 1.020 (0.755, 1.378) | 0.897 | 0.897 |
| | IVW (fixed effects) | | 1.020 (0.755, 1.378) | 0.897 | 0.897 |
| | MR Egger | | 1.837 (1.017, 3.317) | 0.059 | 0.354 |
| | Weighted median | | 1.111 (0.722, 1.710) | 0.632 | 0.758 |
| | CML | | 1.003 (0.745, 1.350) | 0.985 | 0.985 |
| AS intake in tea on T2DM (SNPs=16) | IVW (random effects) | | 1.174 (0.933, 1.477) | 0.172 | 0.206 |
| | IVW (fixed effects) | | 1.174 (0.965, 1.428) | 0.108 | 0.129 |
| | MR Egger | | 1.282 (0.757, 2.169) | 0.371 | 0.564 |
| | Weighted median | | 0.970 (0.726, 1.296) | 0.836 | 0.836 |
| | CML | | 1.160 (0.921, 1.462) | 0.208 | 0.249 |

**Fig 3. Genetically predicted causal associations of intake of different sources of AS on T2DM were assessed by different methods.** IVW, Inverse-Variance-Weighted; FDR, False Discovery Rate; OR, odds ratio; CI, confidence interval; AS, artificial sweetener; T2DM, type 2 diabetes; CML, constrained maximum likelihood.

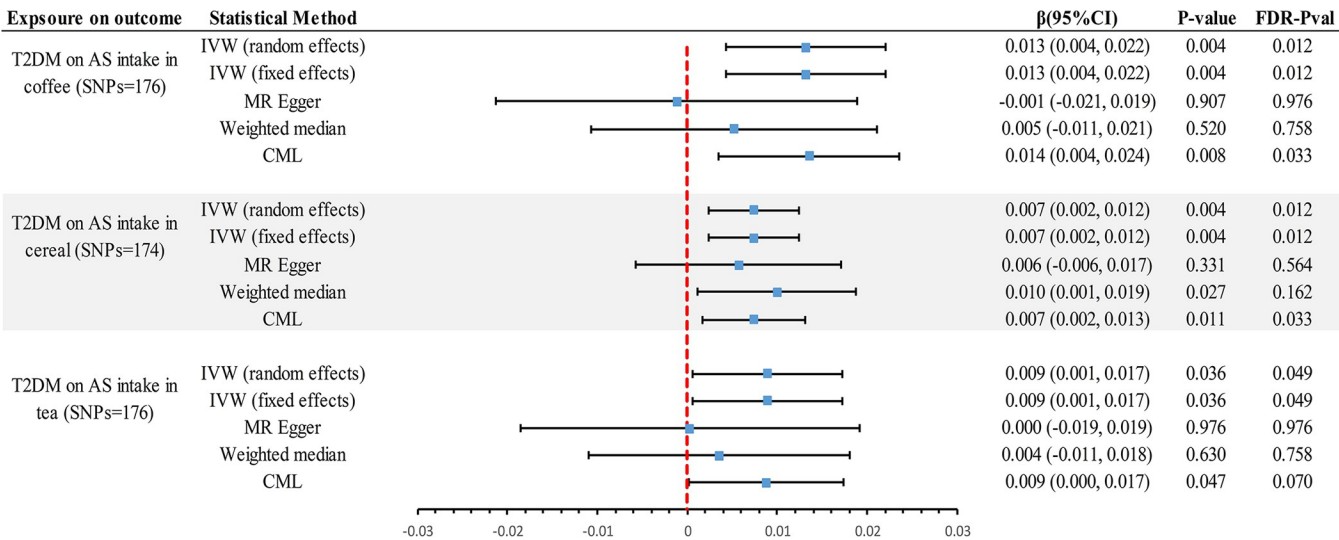

**Fig 4. Genetically predicted causal associations of T2DM on intake of different sources of AS were assessed by different methods.** IVW, Inverse-Variance-Weighted; FDR, False Discovery Rate; β, beta; CI, confidence interval; AS, Artificial sweetener; T2DM, type 2 diabetes; CML, constrained maximum likelihood.

coffee (Mediation effect: 29.50%) and T2DM (**Table 2**). The causal relationships among the three align with the principles of mediation MR ($P > 0.05$).

A series of sensitivity analyses confirmed the robustness of the forward and reverse UVMR results (**Table 3**). Cochran's Q statistic suggested no heterogeneity ($P > 0.05$). MR-PRESSO detected no outliers and no evidence of horizontal pleiotropy ($P > 0.05$). MR-Egger detected no horizontal pleiotropy ($P > 0.05$). The leave-one-out analysis further validated that the causal relationship wasn't influenced by any single SNP (**S1 Fig**), and the funnel plot showed symmetry (**S2 Fig**). The Steiger test indicated that all SNPs passed the test, and the direction of causality remained unchanged, further solidifying the results. The forest plots can be found in **S3 Fig**.

## Discussion

This study conducted a comprehensive MR analysis to delve deeper into the genetic susceptibility linking AS intake from various sources with T2DM. The MR findings corroborated prior epidemiological studies [11–14], establishing a causal relationship between an elevated risk of AS intake in coffee and T2DM. Moreover, we identified a positive correlation between T2DM and AS intake in coffee, cereal, and tea. Further LDSC analysis revealed significant genetic correlation between the exposure and outcome phenotypes. MVMR analyses unveiled

**Table 2. Mediation analysis of the mediation effect of AS intake in coffee on T2DM via confounding factors.**

| Outcome | Mediator | Total effect | Direct effect | Mediation effect | | |
|---|---|---|---|---|---|---|
| | | Effect size (95% CI) | Effect size (95% CI) | Effect size (95% CI) | IE div TE(%) | *P* |
| T2DM | LDL-C | 0.235 (0.035, 0.435) | 0.239 (0.037, 0.440) | -0.004 (-0.028, 0.020) | -1.68% | 0.719 |
| | HDL-C | 0.235 (0.035, 0.435) | 0.166 (-0.044, 0.375) | 0.069 (0.007, 0.132) | 29.50% | 0.026 |
| | TG | 0.235 (0.035, 0.435) | 0.220 (0.016, 0.425) | 0.015 (-0.027, 0.056) | 6.18% | 0.456 |
| | HbA1c | 0.235 (0.035, 0.435) | 0.225 (0.024, 0.425) | 0.010 (-0.004, 0.024) | 4.22% | 0.150 |
| | BMI | 0.235 (0.035, 0.435) | 0.063 (-0.243, 0.369) | 0.172 (-0.059, 0.403) | 73.18% | 0.145 |

BMI, body mass index; IE div TE, Indirect Effect divided by Total Effect; OR, odds ratio; CI, confidence interval; LDL-C, Low Density Lipoprotein Cholesterol; HDL-C, High Density Lipoprotein Cholesterol; TG, Triglyceride; HbA1c, Glycated Hemoglobin A1c.

**Table 3. Summary of sensitivity results.**

| Exposure | Outcome | MR-Egger intercept | | | MR-PRESSO global test | | | Cochrane's Q | | | Steiger_test | |
|---|---|---|---|---|---|---|---|---|---|---|---|---|
| | | Intercept | SE | Pval | RSS$_{obs}$ | P-value | Outlier | Q | Q_df | Q_pval | Direction | Pval |
| AS intake in coffee | T2DM | 0.002 | 0.006 | 0.780 | 16.898 | 0.361 | NA | 13.801 | 13 | 0.388 | TRUE | 7.74E-65 |
| AS intake in cereal | T2DM | -0.011 | 0.005 | 0.036 | 20.690 | 0.480 | NA | 18.773 | 19 | 0.471 | TRUE | 1.09E-84 |
| AS intake in tea | T2DM | -0.003 | 0.007 | 0.719 | 23.803 | 0.150 | NA | 20.726 | 15 | 0.146 | TRUE | 1.12E-73 |
| T2DM | AS intake in coffee | 0.001 | 0.001 | 0.120 | 160.107 | 0.827 | NA | 158.043 | 175 | 0.816 | TRUE | 1.38E-112 |
| T2DM | AS intake in tea | 1.23E-04 | 3.88E-04 | 0.752 | 145.839 | 0.943 | NA | 144.285 | 173 | 0.945 | TRUE | 3.00E-116 |
| T2DM | AS intake in cereal | 0.001 | 0.001 | 0.317 | 133.698 | 0.991 | NA | 132.402 | 173 | 0.990 | TRUE | 7.25E-123 |

AS, Artificial sweetener; T2DM, Type 2 Diabetes Mellitus; MR, Mendelian Randomization; MR-PRESSO, MR Pleiotropy Residual Sum and Outlier.

the influence of several confounding factors, while mediation MR indicated that HDL-C partially mediates the causal relationship.

Previous epidemiological research has observed a link between AS and T2DM. The findings demonstrated an association between artificial sweetener usage and the emergence of insulin resistance and T2DM among diabetic patients, leading to a heightened occurrence of obesity. However, animal studies have hinted at a negative relationship between AS and T2DM [40–42]. In contrast, thorough safety assessments have confirmed their safety [43], and reputable organizations have vouched for their safety [7]. Nevertheless, innate limitations in observational studies make it difficult to fully negate the impact of unobserved confounding variables and reverse causality. Observational studies tend to prioritize correlation over causation. By employing MR analysis, this study minimized the effects of bias and confounding factors, establishing a causal relationship between AS intake in coffee and T2DM.

This study elucidates the results through gastrointestinal reactions, insulin resistance and secretion, alterations in the microbiome, and changes in feeding behavior. Chlorogenic acid and caffeic acid, as bioactive components in coffee, are known to influence intestinal motility and gastric acid secretion, subsequently affecting food and nutrient absorption and digestion [44]. When combined with AS, these compounds could potentially influence the secretion of gut hormones. Research conducted by Jing Ma's team suggests that AS can stimulate the secretion of GLP-1 and GIP from the intestinal endocrine cell line GLUTag, as well as GLP-1 secretion from the human L-cell line NCI-H1 [45], subsequently influencing insulin secretion and glucose homeostasis. Furthermore, intake of AS might alter the structure of the gut microbiome, resulting in gut bacterial imbalance and glucose metabolic disturbances. Consistent evidence provided by research from Jotham Suez et al. [46] indicates that AS can modify the gut microbiome to induce glucose intolerance in mice and various human subgroups, resulting in sustained hyperglycemic states. Coffee itself, with its bioactive compounds like caffeine and polyphenols, can also alter the gut flora. A review by Astrid Nehlig [47] indicates that coffee consumption mainly affects the population levels of bifidobacteria.

AS may directly or indirectly influence insulin secretion and function. Research by Cristina Bosetti and colleagues [48] suggests that AS such as saccharin can induce pancreatic cells to release insulin, resulting in short-term hyperinsulinemia. Long-term overstimulation might lead to the functional exhaustion of pancreatic cells. Caffeine can accelerate gastric emptying, temporarily increasing blood glucose and insulin resistance. A meta-analysis involving seven cohorts by Xiuqin Shi's team supports this notion [49], suggesting that caffeine intake can reduce insulin sensitivity in healthy subjects, possibly due to interference with intracellular glucose uptake. Considering both effects, artificial sweeteners in coffee might exacerbate this burden, leading to overexertion of pancreatic cells or further glucose metabolic disruption.

Coffee, an integral part of daily life, is frequently consumed with high-sugar, high-fat foods like pastries, potentially impacting glucose absorption and metabolic rates [50]. Given the "zero-calorie" characteristic of AS, if consumers mistakenly believe that coffee with AS can off-set the intake of other unhealthy foods, this could lead to an overall increase in caloric intake, thus elevating the risk of T2DM. This notion is further corroborated in the reverse MR analysis, where an increased intake of AS from various sources is associated with the T2DM. Other beverages, such as tea, might be consumed independently at other times of the day. Research by Bangde Li and colleagues suggests sweetness and coffee flavor directly influence two key sensory attributes for consumers [51]. The robust flavor of coffee, compared to tea or cereals, might necessitate more artificial sweeteners to achieve the desired sweetness. Consequently, the amount of sweetener consumed might vary, further explaining the study's findings that AS intake from other sources doesn't show a causal link with T2DM.

The mediation MR analysis unveils a pivotal role of HDL-C as an intermediary in the relationship between AS intake in coffee and the risk of T2DM. HDL-C, commonly termed as the 'beneficial cholesterol,' facilitates reverse cholesterol transport by actively sequestering surplus cholesterol from peripheral tissues and conveying it to the liver for subsequent excretion [52]. AS intake in coffee may influence HDL-C levels by altering metabolism and regulating lipid metabolism through changes in the gut microbiome [53]. Ample evidence suggests that artificial sweeteners are associated with liver damage [54, 55], and liver function, in turn, impacts the synthesis and metabolism of HDL-C. Concurrently, HDL-C may affect insulin sensitivity by modulating the function of β-cells and the peripheral tissue's glucose uptake [56]. Therefore, the mediating effect of HDL-C implies that interventions targeting the regulation of HDL-C, combined with controlling AS intake, might offer a synergistic approach for preventing or mitigating the risk of T2DM.

The study demonstrates several notable merits. Primarily, this MR analysis pioneers in establishing a causal linkage between sources of AS intake and T2DM. Furthermore, given that all SNPs utilized as IVs were identified within the European population, the probability of population stratification bias is diminished, thereby bolstering the credibility of the MR assumption. In the course of this inquiry, the application of rigorous instruments (e.g., F statistic significantly surpassing 10) serves to mitigate potential biases stemming from sample overlap [57]. However, our study is not without certain limitations. While every SNP was scrutinized, not all potential risk factors were considered. Furthermore, the selection of a relatively small number of SNPs as IVs could account for a minimal percentage of exposure variation, thereby affecting the statistical power of causal estimations. In addition, the lack of extensive disease severity and demographic information in the GWAS database, making it impossible to undertake further subgroup analyses.

## Conclusion

In summary, the MR analysis has established a causal relationship between AS intake in coffee and an elevated risk of T2DM, with HDL-C mediating a portion of this causal effect. The reverse analysis indicates a positive correlation between T2DM and artificial sweetener intake from all sources. Future MR analyses, employing larger-scale GWAS summary data and an increased number of genetic instruments, are necessary to corroborate the conclusions drawn from this study.

## Supporting information

**S1 Table. Power calculations for bidirectional univariable Mendelian randomization analyses.**
(DOCX)

**S2 Table. Summary of UVMR analysis results.**
(DOCX)

**S3 Table. Summary of analytical results for MVMR.**
(DOCX)

**S1 Fig. Funnel plot of instrument precision against instrumental variable estimates for each genetic variant separately for Mendelian randomization analysis of exposure on outcomes risk.** (A) AS intake in coffee on T2DM (B) AS intake in tea on T2DM (C) AS intake in cereal on T2DM (D) T2DM on AS intake in coffee (E) T2DM on AS intake in tea (F) T2DM on AS intake in cereal. Solid vertical line is the (random-effect) inverse-variance weighted estimate.
(DOCX)

**S2 Fig. Leave-one-out plot for MR analysis of exposure on outcomes risk.** (A) AS intake in coffee on T2DM (B) AS intake in tea on T2DM (C) AS intake in cereal on T2DM (D) T2DM on AS intake in coffee (E) T2DM on AS intake in tea (F) T2DM on AS intake in cereal.
(DOCX)

**S3 Fig. Single-SNP analysis forest plots of the effect of exposure on outcomes.** (A) AS intake in coffee on T2DM (B) AS intake in tea on T2DM (C) AS intake in cereal on T2DM (D) T2DM on AS intake in coffee (E) T2DM on AS intake in tea (F) T2DM on AS intake in cereal. Point estimates represent the variant-specific ratio estimates for each SNP (in black), and the inverse-variance weighted (IVW) estimate (in orange). Horizontal lines represent 95% confidence intervals around the variant-specific ratio estimates and the IVW estimate.
(DOCX)

## Acknowledgments

Exposure phenotype summary-level data were obtained from ieu open gwas project (https://gwas.mrcieu.ac.uk/) and UK Biobank (http://www.nealelab.is/uk-), and outcomes were obtained from DIAbetes Genetics Replication And Meta-analysis (http://www.nealelab.is/uk-). biobank), endpoints from DIAbetes Genetics Replication And Meta-analysis (https://www.diagram-consortium.org/about.html), and mediator phenotypes from Global Lipids Genetics Consortium, Meta-Analyses of Glucose and Insulin-related traits Consortium, Genetic Investigation of Anthropometric Traits, respectively. The authors would like to thank all researchers for sharing these data.

## Author Contributions

**Conceptualization:** Youqian Zhang, Zitian Tang.

**Data curation:** Youqian Zhang.

**Formal analysis:** Youqian Zhang.

**Investigation:** Youqian Zhang.

**Methodology:** Youqian Zhang.

**Resources:** Yong Shi.

**Software:** Youqian Zhang, Lin Li.

**Supervision:** Lin Li.

**Validation:** Zitian Tang.

**Writing – original draft:** Youqian Zhang.

**Writing – review & editing:** Youqian Zhang, Lin Li.

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
