## [Decision Letter · Decision Letter 0]

4 Oct 2023

PONE-D-23-17352Associations Between Artificial Sweetener Intake from Cereals, Coffee, and Tea and the Risk of Type 2 Diabetes: A Two-Sample Mendelian Randomization AnalysisPLOS ONE

Dear Dr. Li,

Thank you for submitting your manuscript to PLOS ONE. After careful consideration, we feel that it has merit but does not fully meet PLOS ONE’s publication criteria as it currently stands. Therefore, we invite you to submit a revised version of the manuscript that addresses all the points raised by the reviewer during the review process.

We look forward to receiving your revised manuscript.

Kind regards,

Pratibha V. Nerurkar, Ph.D

Academic Editor

PLOS ONE

4. PLOS requires an ORCID iD for the corresponding author in Editorial Manager on papers submitted after December 6th, 2016. Please ensure that you have an ORCID iD and that it is validated in Editorial Manager. To do this, go to ‘Update my Information’ (in the upper left-hand corner of the main menu), and click on the Fetch/Validate link next to the ORCID field. This will take you to the ORCID site and allow you to create a new iD or authenticate a pre-existing iD in Editorial Manager. Please see the following video for instructions on linking an ORCID iD to your Editorial Manager account: ". " ext-link-type="uri" xlink:type="simple">https://www.youtube.com/watch?v=_xcclfuvtxQ"". 

5. Please amend your authorship list in your manuscript file to include author Yong Shi.

6. We note you have included a table to which you do not refer in the text of your manuscript. Please ensure that you refer to Table S2 in your text; if accepted, production will need this reference to link the reader to the Table.

Reviewers' comments:

Reviewer's Responses to Questions

**Comments to the Author**

1. Is the manuscript technically sound, and do the data support the conclusions?

Reviewer #1: Yes

2. Has the statistical analysis been performed appropriately and rigorously? 

Reviewer #1: Yes

3. Have the authors made all data underlying the findings in their manuscript fully available?

Reviewer #1: Yes

4. Is the manuscript presented in an intelligible fashion and written in standard English?

Reviewer #1: Yes

5. Review Comments to the Author

Reviewer #1: MR takes into account genetic variations using GWAS data as instrumental variables to assess the causal relationship between phenotypes. While MR is robust to unmeasured confounding of the exposure-outcome relationship and measurement error associated with the exposure, identifying the direction of causality is especially difficult in the presence of pleiotropy. Methods to address the impact of pleiotropy often have unverifiable assumption with respect to the number of SNPs that are considered as instrumental variable for exposure. One can never fully rule out reverse causality and the effect that T2DB has on AS consumption. Distinguishing causation from correlation is essential in MR studies. That is, the investigator may never be certain which is the outcome or exposure variable (see PubMed: 31822846). Nonetheless, the manuscript is well written and focused. The research question is clear and the authors carefully note the core independent variable assumptions. As well, they were attentive when selecting genetic variants to be used as instruments and provide reasonable sensitivity analyses. However, one concern that the authors could further address is reproducibility.

6. PLOS authors have the option to publish the peer review history of their article (what does this mean?). If published, this will include your full peer review and any attached files.

Reviewer #1: No

---

## [Author Response · Author response to Decision Letter 0]

13 Oct 2023

Dear Editors and Reviewers:

On behalf of all contributing authors, I would like to sincerely thank you for your letter and the constructive comments provided by the reviewers on our article titled " Associations Between Artificial Sweetener Intake from Cereals, Coffee, and Tea and the Risk of Type 2 Diabetes: A Two-Sample Mendelian Randomization Analysis" (Manuscript ID: PONE-D-23-17352). We highly value these comments, as they have greatly contributed to the enhancement of our manuscript. Based on the suggestions from the associate editor and reviewers, we have made extensive revisions to our draft and added additional data to substantiate our findings. In this revised version, all changes to our manuscript are highlighted in yellow text. Following this letter, we provide point-by-point responses to the comments from the associate editor and the reviewer.

Editor

Response: Thank you for pointing out the style requirements of PLOS ONE. In the revised submission, we have meticulously formatted the manuscript in accordance with the PLOS ONE style templates provided. We have also ensured that the file naming adheres to the journal's guidelines. We appreciate your guidance to ensure our manuscript aligns with the journal's standards.

Response: Thank you for emphasizing the importance of data availability. I'd like to clarify that our study utilizes data from previously published and publicly accessible GWAS research. As our work is a secondary analysis of this existing data, there isn't any new data generated that requires repository storage. Furthermore, we have elaborated on this point in our cover letter and made changes to our Data Availability statement accordingly, to better reflect the nature of our data sourcing and its availability. We appreciate your understanding and assistance in ensuring transparency and adherence to the journal's policies.

Response: Thank you for highlighting PLOS's data availability policy. To reiterate, our study exclusively utilizes data from previously published and publicly accessible GWAS research. As it is a secondary analysis of existing data, there is no new, primary data generated by our study that requires repository storage. We have addressed the Data Availability statement in our revised submission, as well as in our response to the previous concern. The underlying data for our study is derived from sources that are already available to the public. Consequently, there is no additional or unique minimal data set from our study to be deposited. We appreciate your dedication to transparency and data accessibility, and we believe our revised statement and cover letter now align with the journal's standards in this regard.

4. PLOS requires an ORCID iD for the corresponding author in Editorial Manager on papers submitted after December 6th, 2016. Please ensure that you have an ORCID iD and that it is validated in Editorial Manager. To do this, go to ‘Update my Information’ (in the upper left-hand corner of the main menu), and click on the Fetch/Validate link next to the ORCID field. This will take you to the ORCID site and allow you to create a new iD or authenticate a pre-existing iD in Editorial Manager. Please see the following video for instructions on linking an ORCID iD to your Editorial Manager account: https://www.youtube.com/watch?v=_xcclfuvtxQ"".

Response: Thank you for the guidance on ORCID iD requirements. I've now linked the corresponding author's ORCID iD: 0009-0006-3868-1296 to our Editorial Manager account. We appreciate your patience and assistance in ensuring that our manuscript adheres to PLOS's guidelines.

5. Please amend your authorship list in your manuscript file to include author Yong Shi.

Response: Thank you for pointing out the oversight regarding our authorship list. We have amended the manuscript to include "Yong Shi" as one of the authors in our revised submission. We appreciate your diligence in ensuring accurate representation of contributors.

6. We note you have included a table to which you do not refer in the text of your manuscript. Please ensure that you refer to Table S2 in your text; if accepted, production will need this reference to link the reader to the Table.

Response: Thank you for pointing out the oversight regarding Table S2. In our revised manuscript, we have made sure to reference Table S2 within the text. Your attention to detail is much appreciated, and we're grateful for the reminder to ensure completeness.

Response: Thank you for highlighting the importance of reference accuracy. We have diligently reviewed and cross-checked every entry in our reference list for the revised manuscript. We can confirm that all cited papers are current, relevant, and have not been retracted. In case any retracted papers were inadvertently included in our initial submission, they have now been removed or replaced, and any such changes are outlined in our revised document. We appreciate your guidance in maintaining the integrity of our research.

Reviewer #1

1.MR takes into account genetic variations using GWAS data as instrumental variables to assess the causal relationship between phenotypes. While MR is robust to unmeasured confounding of the exposure-outcome relationship and measurement error associated with the exposure, identifying the direction of causality is especially difficult in the presence of pleiotropy. Methods to address the impact of pleiotropy often have unverifiable assumption with respect to the number of SNPs that are considered as instrumental variable for exposure. One can never fully rule out reverse causality and the effect that T2DB has on AS consumption. Distinguishing causation from correlation is essential in MR studies. That is, the investigator may never be certain which is the outcome or exposure variable (see PubMed: 31822846). Nonetheless, the manuscript is well written and focused. The research question is clear and the authors carefully note the core independent variable assumptions. As well, they were attentive when selecting genetic variants to be used as instruments and provide reasonable sensitivity analyses. However, one concern that the authors could further address is reproducibility.

Response: First and foremost, we extend our sincere gratitude for your meticulous review and valuable feedback on our manuscript. Your insights are instrumental in enhancing the quality and rigor of our work. Addressing the issue of reproducibility, we wish to clarify that our MR analysis is predicated upon publicly available data that underwent secondary analysis. In our revised manuscript, we have incorporated a "Data Availability" section that elucidates the origins of our data sets. Additionally, Table 1 has been supplemented with a detailed description of our data sources to further enhance clarity and transparency. Furthermore, we would like to highlight that this manuscript was penned in June 2023. Given the progressive nature of the scientific community and in alignment with the journal's publishing requisites, we've undertaken necessary refinements. Specifically, we have enriched our analysis by incorporating MVMR, LDSC, and mediation MR analysis. Notably, for T2DM, we employed the most up-to-date and comprehensive GWAS dataset. Consequently, this led to modifications in our results and discussion sections. The specific changes can be discerned in the subsequent modification list provided. Once again, we deeply appreciate your efforts and guidance, which were pivotal in refining our research paper. We hope that our revisions will meet your expectations and those of the journal.

Revised:

1. The title of the article was revised to read: Associations Between Artificial Sweetener Intake from Cereals, Coffee, and Tea and the Risk of Type 2 Diabetes Mellitus: A Genetic Correlation , Mediation, and Mendelian Randomization Analysis

2. Abstract: Methods added "Linkage disequilibrium score (LDSC) regression was used to evaluate the correlation between phenotypes. Multivariate and mediation MR were applied to investigate confounding factors and mediating effects. Data on AS intake from different sources (N=64, 949) were sourced from the UK. 949) were sourced from the UK Biobank, while T2DM data were derived from the DIAbetes Genetics Replication And Meta-analysis.", the results section for the analysis was reworded. The MR analysis has established a causal relationship between AS intake in coffee and T2DM. The mediation by HDL-C emphasizes potential metabolic pathways that may lead to the development of a new, more efficient, and more effective healthcare system. emphasizes potential metabolic pathways underpinning these relationships".

3. Introduction: the last sentence was revised to read "This study aims to employ univariate MR (UVMR), multivariate MR (MVMR), mediation MR, and linkage disequilibrium score (LDSC) regression to investigate the relationship between intake of AS from various sources and T2DM, further delving into the relationship between intake of AS from various sources and T2DM. disequilibrium score (LDSC) regression to investigate the relationship between intake of AS from various sources and T2DM, further delving into the mediating roles of five confounding factors. mediating roles of five confounding factors. "Pointing out the main theme

4. Methods:

(1) The first section revisits the three core assumptions of Mendelian Randomization (MR), highlights ethical principles, and includes supplementary Data Source Table 1.

(2) Selection of genetic instrumental variables: The methods for gathering information on AS intake from different sources were detailed as follows: "This information was collected using questionnaires where participants detailed the amount of AS (e.g., Canderel) they added to their daily coffee or tea/infusion on a per-drink basis. Moreover, individuals who reported consuming cereal or porridge the previous day specified the amount of sweetener added per bowl." Additionally, the selection criteria for proxy SNPs and alleles were expanded upon.

(3) Source of outcome phenotypes: An added description about the source of T2DM is as follows: "The summary-level GWAS meta-analysis for T2DM integrated 22 cohorts, sourced from the AMP-T2D Knowledge Portal and the DIAbetes Genetics Replication And Meta-analysis (DIAGRAM) consortium[24]. T2DM is defined by ICD-10 codes, encompassing 180,834 cases and 492,191 controls from European populations."

(4) There's an added Data sources for possible mediators section used for MVMR and mediation MR analysis.

5. Statistical analyses: The section is now organized into Primary MR analysis, Genetic Correlation Analysis, and Sensitivity analysis. Considering multiple comparisons, an FDR correction has been implemented. To enhance data reliability, constrained maximum likelihood (CML) was introduced. Additionally, Mediation MR and MVMR analysis and Linkage disequilibrium score (LDSC) were employed to bolster causality assertions. Sensitivity analysis now incorporates the genetic variation explained (R2) and POWER calculations.

6. Results

(1) An addition notes that "LDSC revealed a significant genetic correlation between all exposures and outcome phenotypes."

(2) Within the forward MR analysis, it was found that " Specifically, for every standard deviation (SD) increase in genetically predicted AS intake in coffee, the risk of T2DM increased by 26% (OR = 1.265, 95% CI 1.035~1.545, P = 0.021, PFDR = 0.042).” In the reverse MR analysis utilizing the IVW method (Fig. 4), there is compelling evidence suggesting a positive causal relationship between T2DM and AS intake in coffee (β = 0.013, 95% CI 0.004~0.022, P = 0.004, PFDR = 0.012), cereal (β = 0.007, 95% CI 0.002~0.012, P = 0.004, PFDR = 0.012), and tea (β = 0.009, 95% CI 0.001~0.017, P = 0.036, PFDR = 0.049)

(3) Subsequent mediation MR disclosed that "HDL-C partially mediates the causal link between AS intake in coffee and T2DM, accounting for a mediation effect of 29.50%

7.Discussion:

(1) The opening paragraph was refined to: "This study executed a comprehensive MR analysis to delve into the genetic predisposition connecting AS intake from diverse sources with T2DM. Our MR findings are consistent with previous epidemiological studies[11–14], reinforcing a causal association between increased AS intake in coffee and T2DM. Furthermore, we pinpointed a positive correlation between T2DM and AS intake in coffee, cereal, and tea. Subsequent LDSC analysis uncovered a significant genetic correlation between exposure and outcome phenotypes. MVMR analyses highlighted the potential confounding factors, while mediation MR suggested HDL-C's partial mediating role in the relationship."

(2) The elucidation of the positive findings was expanded: "This study interprets the findings through several mechanisms including gastrointestinal reactions, insulin resistance and secretion, alterations in the gut microbiome, and shifts in feeding behavior."

(3) A more in-depth analysis of the mediating factor, HDL-C, was added: "The mediation MR unveils HDL-C as a linchpin in the nexus between AS intake in coffee and T2DM risk. Commonly dubbed 'good cholesterol,' HDL-C facilitates the reverse cholesterol transport, actively scavenging excess cholesterol from peripheral tissues and shuttling it to the liver for subsequent excretion[53]. Coffee's AS intake might modulate HDL-C levels by altering metabolism and dictating lipid metabolism via shifts in the gut microbiome[54]. There's substantial evidence aligning artificial sweeteners with liver detriment[55,56], and liver function reciprocally impacts HDL-C synthesis and metabolism. Concurrently, HDL-C can influence insulin sensitivity by modulating β-cell functionality and glucose uptake in peripheral tissues[57]. Thus, HDL-C's mediation underscores that targeting HDL-C regulation, conjoined with AS intake moderation, could offer a two-pronged strategy to preempt or mitigate T2DM risk.

---

## [Editor Report · Decision Letter 1]

16 Oct 2023

Associations Between Artificial Sweetener Intake from Cereals, Coffee, and Tea and the Risk of Type 2 Diabetes Mellitus: A Genetic Correlation, Mediation, and Mendelian Randomization Analysis

PONE-D-23-17352R1

Dear Dr. Li,

We’re pleased to inform you that your manuscript has been judged scientifically suitable for publication and will be formally accepted for publication once it meets all outstanding technical requirements.

Kind regards,

Pratibha V. Nerurkar, Ph.D

Academic Editor

PLOS ONE
---

## [Editor Report · Acceptance letter]

18 Oct 2023

PONE-D-23-17352R1 

Associations Between Artificial Sweetener Intake from Cereals, Coffee, and Tea and the Risk of Type 2 Diabetes Mellitus: A Genetic Correlation, Mediation, and Mendelian Randomization Analysis 

Dear Dr. Li:

I'm pleased to inform you that your manuscript has been deemed suitable for publication in PLOS ONE. Congratulations! Your manuscript is now with our production department. 

Kind regards, 

on behalf of

Dr. Pratibha V. Nerurkar 

Academic Editor

PLOS ONE